# Bio-Organic Fertilizer Application Enhances Silage Maize Yield by Regulating Soil Physicochemical and Microbial Properties

**DOI:** 10.3390/microorganisms13050959

**Published:** 2025-04-23

**Authors:** Ying Tang, Lili Nian, Xu Zhao, Juan Li, Zining Wang, Liuwen Dong

**Affiliations:** Institute of Soil, Fertilizer and Water-Saving Agriculture, Gansu Academy of Agricultural Sciences, Lanzhou 730070, China; nll18893814845@163.com (L.N.); zhaoxu6939438@163.com (X.Z.); lijuanh@126.com (J.L.); 17722668551@163.com (Z.W.); dongliuwen2020@163.com (L.D.)

**Keywords:** bio-organic fertilizer, silage maize yield, microbial diversity, ecosystem multifunctionality

## Abstract

Silage maize is vital to livestock development in northern China, but intensive chemical fertilization has led to soil degradation and reduced productivity. Bio-organic fertilizers offer a sustainable alternative, though their effects on soil multifunctionality remain underexplored. This study evaluated the impact of combining decomposed cow manure, *Bacillus amyloliquefaciens*, and mineral potassium fulvic acid with chemical fertilizers (NPK) on silage maize yield, soil microbial diversity, and ecosystem multifunctionality (EMF). Field experiments showed that bio-organic fertilization increased silage maize yield by 10.23% compared to chemical fertilizers alone, primarily by boosting labile organic carbon and soil enzyme activity. It also enhanced bacterial richness and diversity, with little effect on fungal communities. Microbial network analysis revealed more complex and stable bacterial networks under bio-organic treatments, indicating strengthened microbial interactions. Random forest and structural equation modeling (SEM) identified soil carbon storage and bacterial diversity as key drivers of EMF, which integrates soil functions such as nutrient cycling, decomposition, enzyme activity, and microbial diversity. These findings suggest that soil bacterial diversity and its interactions with soil properties are critical to both crop productivity and soil health. The optimal fertilization strategy for silage maize in this region involves the combined use of cattle manure, *Bacillus amyloliquefaciens*, mineral potassium fulvic acid, and NPK fertilizers. This approach improves yield, microbial diversity, and soil multifunctionality. Future studies should consider environmental variables and crop varieties across diverse regions to support broader application.

## 1. Introduction

Silage maize (*Zea mays*) is widely cultivated worldwide due to its favorable traits and significant role in animal husbandry. In the United States, for instance, the annual cultivation area of silage maize exceeds 2.4 million hectares, accounting for more than 7% of the total maize area [1]. In recent years, China’s implementation of the “grain-for-feed” strategy has significantly increased the planting area of silage maize, reaching 1.67 million hectares in 2020 [2]. Silage maize offers several advantages, including high energy content and favorable silage characteristics. Furthermore, studies have shown that incorporating corn silage into the diets of dairy cows, alongside grass or grass silage, enhances feed intake, milk yield, and milk protein content [3]. As a result, silage maize has become a crucial roughage source for ruminants such as dairy cows, beef cattle, and sheep. Zhangye City in northwestern China is a typical oasis agricultural region, a major grain production base, and a significant area for corn cultivation [4]. In recent years, with the adjustment and optimization of the agricultural industrial structure, animal husbandry has emerged as the leading sector of local agriculture. The scale of livestock breeding has expanded annually, with continuous improvements in standardized production and industrialized management. As a result, the district has become a key cattle and sheep breeding hub in China [5]. The supply of high-quality forage is crucial for the sustainable development of this industry. Studies have demonstrated that silage maize offers several advantages, including high biomass, excellent nutritional value, good palatability, and low feeding costs [1]. The development of silage corn presents a critical solution to the challenge of supporting the rapid growth of animal husbandry while addressing forage shortages.

In the past few decades, the widespread use of chemical fertilizers has significantly boosted crop yields. However, the long-term overuse of chemical fertilizers has not only led to adverse environmental effects, such as water and air pollution, but has also contributed to soil degradation, negatively impacting the health and function of agricultural ecosystems [6]. As a potential alternative, organic fertilizers can reduce dependence on chemical fertilizers and help mitigate soil degradation [7]. However, when applied alone, organic fertilizers may limit crop growth due to their slow nutrient release rate and prolonged release period [8]. Biostimulants and functional bacterial powders can accelerate the nutrient release process of organic fertilizers, thereby addressing this limitation. Therefore, the combined application of chemical fertilizers, organic fertilizers, biostimulants, and functional bacterial powders can maximize the utilization of soil nutrient resources, improve soil physical and chemical properties, and enhance soil microbial activity. This integrated fertilization strategy is considered a crucial approach for promoting sustainable agricultural development. In this study, we have specifically selected decomposed cow manure, mineral potassium fulvic acid, and *Bacillus amyloliquefaciens* as a combination to improve soil fertility, microbial diversity, and crop yield. Cow manure is rich in essential nutrients like nitrogen, phosphorus, and potassium, and contributes organic matter that improves soil structure and microbial activity, thus promoting nutrient cycling [9]. Mineral humic acid potassium, a combination of organic and mineral compounds, has been shown to improve soil physical properties, increase nutrient availability, and enhance microbial diversity [10]. Studies have demonstrated that mineral humic acid potassium boosts soil fertility and supports beneficial microbial communities, resulting in improved crop productivity [11]. *Bacillus amyloliquefaciens* was selected for its proven ability to enhance plant growth by producing phytohormones, boosting nutrient uptake, and suppressing pathogens. This bacterium accelerates the decomposition of organic matter and mineralizes nutrients, improving the efficiency of slow-release organic fertilizers [12,13]. The novelty of this study lies in the synergistic application of decomposed cow manure, mineral humic acid potassium, and *Bacillus amyloliquefaciens*. While previous research has examined the effects of individual components like organic fertilizers or microbial inoculants on soil health and crop yield, few studies have explored the combined effects of these three components. Our study builds on this existing research by exploring the potential synergistic effects of this unique combination on soil health, microbial diversity, and crop yield, an area that remains underexplored.

Soil microbial communities play a crucial role in soil material cycling and nutrient transformation, serving as key drivers of agricultural productivity [14]. Research has demonstrated that long-term application of chemical fertilizers can alter the structure of soil microbial communities, significantly reducing their diversity [15,16]. These changes not only compromise soil ecological functions but may also hinder the healthy growth of crops. In contrast, the long-term application of a combination of inorganic and organic fertilizers has been shown to effectively restore microbial diversity, potentially returning it to pre-chemical fertilizer levels [17]. Furthermore, this combined fertilization strategy has the added benefit of increasing crop yields, suggesting that soil microorganisms may be a critical factor driving the potential advantages of combining inorganic and organic fertilizers for crop production [18]. Organic fertilizers have a significant effect on the composition of soil microbial communities, thereby regulating carbon and nitrogen cycles in soil. These processes are driven by soil microbial communities and are essential for maintaining soil quality and supporting crop growth. Hu et al. [18] showed that organic fertilizer application significantly increased the relative abundance of microorganisms carrying carbon (C) and nitrogen (N) cycle genes. The research also found that organic fertilizers can promote nitrogen transformation processes and enhance bacterial activity [19]. Through positive effects on soil quality parameters or by changing the structure of soil microbial communities, organic fertilizers can help suppress crop diseases and maintain the productivity of wheat or corn for a long time [20,21]. Liu et al. further showed that organic fertilizer substitution can increase corn yield by regulating the composition of soil biological communities and their functional groups [22]. In addition, organic fertilizer substitution can alleviate the adverse effects of long-term chemical fertilizer application on soil quality by changing soil microbial activity and related soil processes [23], thereby improving soil fertility and regulating crop yields. These research results highlight the potential of organic fertilizers to promote soil ecological functions by optimizing nutrient management.

Soil ecological multifunctionality (EMF) is typically assessed through extracellular enzyme activities, particularly those involved in carbon (C) and nitrogen (N) cycles, offering a quantitative measure of soil biodiversity and ecosystem function [24]. Soil enzyme activity is widely recognized as an indicator of soil quality and ecosystem functional potential due to its sensitivity to long-term fertilization practices [25]. Generally, excessive use of chemical fertilizers inhibits the activity of enzymes related to the carbon cycle, likely because microorganisms no longer depend on enzymatic decomposition to acquire these nutrients [26]. In contrast, organic fertilization alters soil microbial communities and their diversity by influencing the dynamics of dissolved organic carbon (DOC) and microbial biomass carbon (MBC) [27]. Studies have indicated a positive correlation between soil carbon content and microbial diversity [28]. However, Yang et al. found that high nitrogen fertilizer application weakened the relationship between soil carbon content and microbial diversity [2]. Additionally, organic fertilizer application and other crop management practices, such as land tillage, significantly influence carbon metabolism in soil [17]. Therefore, exploring the effects of different fertilization systems on the relationship between soil quality, microbial diversity, and soil ecological multifunctionality is of considerable research importance. This study aims to elucidate the relationship between soil microbial communities, ecosystem multifunctionality, and crop yield in silage maize fields, as well as their responses to various organic fertilizer combinations. To achieve this, the following hypotheses were tested: (1) the application of bio-organic fertilizer, specifically a combination of cow manure, mineral fulvic acid potassium, and *Bacillus amyloliquefaciens*, can improve the community diversity of soil bacteria and fungi in silage maize fields; (2) the improvement in microbial diversity will positively influence soil ecosystem multifunctionality, including nutrient cycling, and will support the growth and productivity of silage maize; and (3) the combined application of these fertilizers will improve nutrient availability and promote silage maize growth by optimizing soil properties and microbial activity, leading to higher yields compared to conventional fertilization methods.

## 2. Materials and Methods

### 2.1. Experimental Design and Soil Sampling

The field experiment was conducted at the farm of Gansu Huarui Agriculture Co., Ltd. in Minle County, Zhangye City, Gansu Province (N 38°73′, E 100°40’), at an elevation of 1676 m. The region has an average annual temperature of 4.1 °C, a frost-free period of 140 days, an average annual evaporation rate of 2075 mm, and annual precipitation of 351 mm. The soil type was irrigated desert soils (Chinese Soil Classification System) or arenosols (FAO Soil Classification System), with the following basic physical and chemical properties: organic matter content of 20.3 g/kg, available nitrogen of 75.8 mg/kg, available phosphorus of 107.5 mg/kg, available potassium of 312.5 mg/kg, and a pH of 7.7.

Urea (N ≥ 46.0%), superphosphate (P_2_O_5_ ≥ 42%), and potassium sulfate (K_2_O ≥ 51%) were used as sources of nitrogen, phosphorus, and potassium fertilizers, respectively. The experiment involved five distinct fertilization treatments: (1) CK, chemical NPK fertilizers (N: 450 kg/ha; K_2_O: 105 kg/ha; P_2_O_5_: 225 kg/ha); (2) T1, chemical NPK fertilizers (10% nitrogen reduction, N: 405 kg/ha; K_2_O: 105 kg/ha; P_2_O_5_: 225 kg/ha) combined with well-decomposed cow manure (3600 kg/ha); (3) T2, chemical NPK fertilizers (10% nitrogen reduction, N: 450 kg/ha; K_2_O: 105 kg/ha; P_2_O_5_: 225 kg/ha) combined with decomposed cow manure (3600 kg/ha) and *Bacillus amyloliquefaciens*, (4) T3, chemical NPK fertilizers (10% nitrogen reduction, N: 450 kg/ha; K_2_O: 105 kg/ha; P_2_O_5_: 225 kg/ha) combined with decomposed cow manure (3600 kg/ha) and mineral potassium fulvic acid (7.2 kg/ha); and (5) T4, chemical NPK fertilizers (10% nitrogen reduction, N: 450 kg/ha; K_2_O: 105 kg/ha; P_2_O_5_: 225 kg/ha) combined with decomposed cow manure (3600 kg/ha), *Bacillus amyloliquefaciens* and mineral potassium fulvic acid (7.2 kg/ha). A wide-narrow row planting pattern was adopted for planting corn (variety Jinling 67). Compared to traditional equal-row planting, the wide-narrow row planting pattern has significant advantages, including improved ventilation and light transmission, optimized soil structure, reduced pests and diseases, and enhanced field management efficiency. These benefits contribute to increased photosynthetic efficiency and corn yield. The film width is 1.8 m, and the film surface is 1.6 m, with corn planted in 4 rows on the film. The row spacing is set at 70 cm for the wide row and 25 cm for the narrow row, with a plant spacing of 19 cm, resulting in a planting density of 105,270 plants per hectare. Drip irrigation was employed, with 10 irrigations and 8 fertilizations throughout the growth period. Phosphate fertilizer was applied at a rate of 225 kg/ha (P_2_O_5_), corresponding to a physical application of 522 kg/ha of heavy superphosphate. A 15% nitrogen fertilizer, a total phosphorus fertilizer, and a bio-organic fertilizer are used as base fertilizers and applied in combination with land preparation before sowing. The remaining 85% nitrogen fertilizer and all potassium fertilizers are applied by drip irrigation. The specific fertilization plan is detailed in Appendix A.

The experiment used a randomized block design with an organic fertilizer application rate of 3600 kg/ha. Each treatment was applied to a 1500 m^2^ plot (30 m × 50 m), and large-area planting was conducted without replication. To account for variability, three random sampling points were selected along a north–south transect within each treatment area. Soil samples were collected at a depth of 20 cm, with five soil cores taken from each sampling point and combined to form a composite sample. Each composite sample was then divided into three portions. One portion was transferred to a sterilized centrifuge tube and immediately placed in an ice box with ice packs. It was then transported to the laboratory and stored in a −80 °C freezer for the extraction of total soil DNA. The second portion was placed in a sealed Ziplock bag, stored in an icebox, and transported to the laboratory under low-temperature conditions. It was subsequently stored at 4 °C for the determination of soil moisture content, enzyme activity, and microbial biomass carbon. The remaining portion was placed in a well-ventilated area to air-dry naturally for the determination of soil pH, active organic carbon components, humus, and other relevant parameters.

### 2.2. Analysis of Soil Physicochemical Properties

Soil pH was measured using a pH meter (Mettler Toledo, Greifensee, Switzerland) with a soil-to-water ratio of 1:2.5. The air-dried soil was ground and sieved through a 2 mm sieve, and an extract was prepared using a soil-to-water ratio of 5:1. The mixture was shaken for 5 min and allowed to stand for 30 min. Soil salinity was determined using the drying method [29]. Available phosphorus (AP) was extracted with 0.5 M NaHCO_3_ (pH = 8.5) and determined using the molybdenum blue method [30]. The concentration of NO_3_^−^-N and NH_4_^+^-N in soils was determined by extraction with KCl and measurement with a continuous flow [31]. Available potassium (AK) content was determined by NH_4_OAc leaching and flame photometry [32]. Soil organic carbon (SOC) was determined using the potassium dichromate method [33]. DOC was extracted using a 0.5 M K_2_SO_4_ solution, with 10 grams of soil shaken in 50 mL of solution for 30 min, followed by filtration and measurement using a TOC analyzer [34]. The easily oxidizable organic carbon (EOC) content in the soil samples was measured using the 333 mol/L KMnO_4_ oxidation-colorimetric method [35]. Microbial biomass carbon was determined using the chloroform fumigation and leaching method [36]. The contents of humic acid (HA), fulvic acid (FA), and humin (HM) were determined using the method described by Liu et al. [37].

Soil enzyme activity was assessed by referring to soil enzymes and their research methods [38]. The activity of soil alkaline phosphatase (S-ALP) was measured at 37 °C using p-nitrophenyl phosphate (PNPP; Sigma N4645, Sigma-Aldrich, St. Louis, MO, USA) as the substrate at pH 11.0 and expressed as μg PNP per hour per gram of dry soil. Catalase (S-CAT) activity was measured using the KMnO_4_ titration method and expressed as milliliters of 0.1 mol/L KMnO_4_ consumed per gram of soil per hour. Cellulase (S-CL) activity was quantified colorimetrically using the 3,5-dinitrosalicylic acid (DNS) method and expressed as the amount of glucose generated per gram of sample through cellulase action. Urease (S-UE) activity was determined using the phenol-sodium hypochlorite colorimetric method and expressed as milligrams of NH_3_-N produced per gram of soil after one day of incubation. Sucrase (S-SC) activity was measured using the 3,5-dinitrosalicylic acid colorimetric method, with sucrase activity expressed as milligrams of glucose produced per gram of soil after one day of incubation.

### 2.3. DNA Extraction and High-Throughput Sequencing

Total microbial genomic DNA was extracted from soil samples using the E.Z.N.A.^®^ Soil DNA Kit (Omega Bio-tek, Norcross, GA, USA) following the manufacturer’s instructions. The quality and concentration of DNA were assessed using 1.0% agarose gel electrophoresis and a NanoDrop2000 spectrophotometer (Thermo Scientific, Waltham, MA, USA) and stored at −80 °C until further use. The hypervariable V3-V4 region of the bacterial 16S rRNA gene was amplified using primer pairs 338F (5′-ACTCCTACGGGAGGCAGCAG-3′) and 806R (5′-GGACTACHVGGGTWTCTAAT-3′) by a T100 Thermal Cycler (BIO-RAD, Hercules, CA, USA). Fungal ITS rRNA gene fragments were amplified using the general fungal primers ITS1F (5′-CTTGGTCATTTAGAGGAAGTAA-3′) and ITS1R (5′-GCTGCGTTCTTCATCGATGC-3′). PCR products were extracted from 2% agarose gels, purified using the PCR Clean-Up Kit (YuHua, Shanghai, China), and quantified using a Qubit 4.0 fluorometer (Thermo Fisher Scientific, USA). Purified amplicons were pooled in equimolar amounts and sequenced using paired-end sequencing on an Illumina NextSeq2000 platform (Illumina, San Diego, CA, USA) following standard protocols by Majorbio Bio-Pharm Technology Co., Ltd. (Shanghai, China). The raw sequencing reads were deposited in the NCBI Sequence Read Archive (SRA) database (Accession Number: PRJNA1234875).

### 2.4. Determination of Ecosystem Multifunctionality

In this study, 15 ecosystem function (EF) indicators were measured and categorized into two groups: (1) functions related to soil carbon nutrients and storage (EF-C), which included soil organic carbon, microbial carbon, easily oxidizable organic carbon, and soluble organic carbon and (2) soil ecosystem multifunctionality (EMF), encompassing all 15 ecosystem function indicators. These indicators were selected because they play a critical role in regulating and maintaining key ecological processes in cornfield ecosystems. Additionally, they are widely used in ecosystem function and multifunctionality research [24,39,40]. The ecosystem multifunctional index was calculated using the mean value method [41]. First, the 15 ecological function indicators were standardized using the formula $f_{ij}=(X_{ij}-{min}_{ij})/({max}_{ij}-{min}_{ij})$. Here, $f_{ij}$ represents the standardized value of the j-th ecosystem function variable for plot i, $X_{ij}$ is the actual measured value of the j-th ecosystem function variable for plot i, ${min}_{ij}$ is the minimum value of the j-th ecosystem function across all plots for the same factor, and ${max}_{ij}$ is the maximum value of the j-th ecosystem function variable across all plots for the same factor.

The individual function method was used to calculate the ecosystem function index (EF) as follows:$${EF}_{ij}=\sum_{j}^{n}\frac{f_{ij}}{n}$$

The average method is used to calculate the ecosystem multifunctionality index EMF:$${EMF}_{i}=\frac{1}{N}\sum_{1}^{N}f_{ij}$$
where ${EF}_{ij}$ is the functional index of the j-th function of plot i. n is the number of ecosystem variable indicators included in this function. ${EMF}_{i}$ is the ecosystem’s multifunctionality index of plot i, calculated as the standardized average of all variable indicators for the plot. N is the total number of ecosystem functions in plot i.

### 2.5. Statistical Analysis

The physical, chemical, and biological properties of soil were analyzed using one-way analysis of variance (ANOVA), with mean separation performed in SPSS 20.0 (IBM SPSS Statistics, Chicago, IL, USA) using Tukey’s multiple comparison test. Differences with *p* < 0.05 were considered statistically significant. Data visualization was conducted with OriginPro 9.1 (OriginLab, Northampton, MA, USA). Principal component analysis (PCA) and network analysis were performed using R (version 4.2.2), while redundancy analysis was carried out with Canoco5. Gephi (version 0.9.2) was used for network visualization. Random forest analyses were conducted with the ‘Random Forest’ package in R to identify key bacterial and fungal predictors of soil function. Structural equation modeling (SEM) was applied to examine the relationships between soil organic carbon (SOC) and microbiological properties using AMOS 22.0 (IBM SPSS Statistics, Chicago, IL, USA). The best-fit model was determined using generalized least squares, with model fitness assessed by a non-significant chi-square test (χ^2^), comparative fit index (CFI), and root mean square error of approximation (RMSEA).

## 3. Results

### 3.1. Maize Yield and Soil Physicochemical Factors

Silage maize yield and soil physicochemical properties were affected by different organic fertilization treatments (Figure 1). The T1, T2, T3, and T4 treatments significantly increased the maize yield by 6.53%, 5.38%, 7.28%, and 10.23%, respectively, compared with CK. The pH values and total salt (TS) content of T1, T2, and T3 were significantly higher than those of CK and T4. The T4 treatment significantly increased the soil organic carbon (SOC) content by 8.03%, while T1, T2, and T3 only increased this by 2.78%, 2.97%, and 3.47%, respectively, compared with CK. The microbial biomass carbon (MBC) content under T1, T2, T3, and T4 treatments increased by 11.52%, 29.90%, 18.40%, and 35.55%, respectively, compared with CK. Additionally, the easily oxidizable organic carbon (EOC) and dissolved organic carbon (DOC) contents in the T4 treatment increased by 33.97% and 17.54%, respectively, compared with CK. Humic acid (HA), fulvic acid (FA), and humin (HM) contents in T1, T2, T3, and T4 treatments were significantly higher than those in CK, with T4 exhibiting the highest levels. Notably, soil catalase (S-CAT) and soil cellulase (S-CL) activities under organic fertilization treatments were significantly higher than those under CK. Furthermore, T1, T2, T3, and T4 treatments increased soil alkaline phosphatase (S-ALP) activity by 13.55%, 16.34%, 18.99%, and 28.66%, respectively, compared with CK. Similarly, soil urease (S-UE) activity increased by 20.17%, 19.76%, 21.10%, and 34.54%, respectively, under the same treatments. Lastly, soil sucrase (S-SC) activity increased by 18.02%, 39.39%, 45.14%, and 63.64%, respectively, in T1, T2, T3, and T4 treatments compared with CK. Correlation analysis revealed that silage corn yield was negatively associated with soil pH and total salt content, while it showed a significant positive correlation with soil enzyme activity and nutrient content (S2). This suggests that higher pH and salinity levels may inhibit nutrient uptake, thereby reducing crop yield, whereas increased enzyme activity and nutrient availability contribute to improved plant growth and higher yields.

### 3.2. Soil Microbial Structural Composition and Community Distribution

Based on their relative abundance, *Proteobacteria* (26.03~31.02%), *Chloroflexi* (15.61~20.08%), *Actinobacteriota* (13.22~15.05%), *Acidobacteriota* (8.61~12.92%), and *Firmicutes* (8.38~11.52%) were the dominant phyla of the soil bacterial community (Figure 2a). The T3 and T4 treatments increased the relative abundance of *Chloroflexi* while decreasing the relative abundances of *Proteobacteria* and *Firmicutes*. The dominant fungal phyla were *Ascomycota* (50.15~63.24%), *unclassified_k__Fungi* (29.90~43.56%), and *Mortierellomycota* (3.40~4.10%) (Figure 2b). Compared with CK, the four organic fertilizer treatments decreased the relative abundance of *Ascomycota* while increasing the relative abundance of *unclassified_k__Fungi*.

Distributional characteristics of bacterial and fungal community structures in response to organic fertilizer application were analyzed using principal component analysis (PCA). PC1 and PC2 accounted for 60.6% (44.6 and 16%, respectively) of bacterial community variation (Figure 3a) and 57.3% (33.9 and 23.4%, respectively) of fungal community variation (Figure 3b). The bacterial communities in the T3 and T4 treatments were distinct from those in the control (CK), whereas the communities in the T1 and T2 treatments closely resembled those in the CK (Figure 3a). The fungal communities in the CK, T1, T2, and T3 treatments were typically clustered together, whereas the T4 treatment was distinctly separated from the others (Figure 3b).

### 3.3. Soil Microbial Community Diversity and Richness

The Shannon, Simpson, ACE, and Chao indices for bacteria and fungi exhibited distinct responses to various organic fertilization treatments (Figure 4). The Simpson and Shannon indices measure microbial community diversity, whereas the ACE and Chao indices indicate microbial community richness [42]. Compared with CK, all organic fertilizer treatments (T1, T2, T3, T4) significantly increased the Shannon, ACE, and Chao indices of the bacterial community, with the T4 treatment showing the greatest increase, by 2%, 7.60%, and 8.11%, respectively. Additionally, the Simpson index in the organic fertilizer treatments was significantly lower than that in CK. However, no significant differences were observed in the Shannon, Simpson, ACE, and Chao indices of the fungal community among the treatments. These results indicate that organic fertilizer application significantly enhanced the diversity and richness of the bacterial community while having no significant impact on the fungal community.

### 3.4. Soil Microbial Community Co-Occurrence Networks

To explore the response of soil bacterial and fungal communities to organic fertilizer application, ecological networks of bacterial and fungal communities were constructed for all treatments (Figure 5). The results showed that different organic fertilizer treatments significantly altered microbial interactions (Table 1). In the bacterial network, organic fertilizer application increased the number of nodes, links, average degree, and graph density, indicating enhanced network complexity. The T4 treatment exhibited the highest proportion of positive links, suggesting that the T4 treatment greatly promoted bacterial cooperation. For the fungal network, organic fertilizer application increased the proportion of positive connections, enhancing fungal species cooperation. The T4 treatment further increased the network’s average degree and graph density, suggesting greater complexity. Regardless of bacterial or fungal networks, the modularity index of organic fertilizer treatments was higher than that of CK, indicating that organic fertilizer application strengthened microbial resistance to environmental disturbances. Among all treatments, the T4 treatment exhibited the highest modularity index for both bacterial and fungal networks, suggesting the strongest network stability.

### 3.5. Multifunctionality of Soil Ecosystems

Different organic fertilizer combinations had a significant impact on both EF-C and EMF (Figure 6a,b). One-way ANOVA revealed that, compared with CK, organic fertilizer application significantly increased soil EF-C and EMF, with the greatest increase observed in the T4 treatment. Specifically, soil EF-C increased more than sixfold, while soil EMF nearly tripled. Among the organic fertilizer treatments, T4 exhibited the highest EF-C and EMF values, followed by T2 and T3, with T1 showing the lowest. The random forest analysis indicated that soil carbon nutrition and storage function were the primary predictors of ecosystem multifunctionality, followed by bacterial diversity (Figure 6c). Structural equation modeling further revealed that soil carbon nutrition and storage function, bacterial diversity, and fungal diversity had significant direct positive effects on ecosystem multifunctionality, whereas enzyme activity exhibited a significant direct negative effect (Figure 6d).

### 3.6. Correlation Analysis Between Soil Physicochemical Properties and Microorganisms

In maize fields, soil bacteria are more responsive to organic fertilizer application than fungi (Figure 7). Different organic fertilizer combinations have varying effects on bacterial and fungal communities through changes in soil physicochemical properties. Specifically, MBC, EOC, DOC, Ha, HM, S-ALP, S-CL, UE, and S-SC are significantly positively correlated with bacterial community richness, while SOC, MBC, and S-CAT show significant positive correlations with fungal community richness. This indicates that organic fertilizers influence microbial community structure by optimizing soil carbon pools and enzyme activity. However, despite the strong correlations between certain soil physicochemical properties and microbial richness, these properties do not significantly affect bacterial or fungal diversity. This suggests that microbial diversity is likely regulated by more complex environmental factors or biotic interactions rather than solely by soil physicochemical properties.

## 4. Discussion

### 4.1. Bio-Organic Fertilizer Changed Corn Yield by Regulating Soil Carbon Pool and Enzyme Activity

Exogenous application of organic materials is a key approach to increasing soil organic carbon content (SOC), and its effectiveness is regulated by the dynamic balance between soil carbon inputs and outputs [43]. The decomposition rates and transformation processes of different types of organic materials in soil vary, which affects the accumulation and stability of labile organic carbon [44]. Additionally, variations in crop species and root exudates can modulate soil microbial community structure, further influencing soil carbon cycling [45]. Environmental factors, such as temperature, precipitation, and soil physicochemical properties, also play a crucial role in SOC dynamics [24]. In this study, compared to the application of chemical fertilizer alone (CK), organic fertilizer treatments (T1–T4) significantly increased soil microbial biomass carbon (MBC). This increase is likely attributed to the enhanced microbial activity induced by organic carbon source inputs, which promoted the decomposition and metabolism of organic matter, thereby facilitating the more efficient conversion of applied residual carbon to SOC [46]. Furthermore, the combined application of cow manure, *Bacillus amyloliquefaciens,* and mineral potassium fulvic acid significantly increased the contents of dissolved organic carbon (DOC) and easily oxidizable organic carbon (EOC) compared with the separate application of cow manure and *Bacillus amyloliquefaciens* or mineral potassium fulvic acid. The observed effects may be attributed to the combined application of cow manure, *Bacillus amyloliquefaciens,* and mineral potassium fulvic acid, which together provide diverse organic carbon sources and enhance the activity and stability of the soil carbon pool. Cow manure serves as the primary carbon source, offering relatively stable organic matter and partially readily available organic compounds, which improve soil structure and provide a continuous carbon supply [17]. Mineral potassium fulvic acid promotes microbial metabolism and enhances organic carbon utilization [47]. *Bacillus amyloliquefaciens* accelerates organic matter decomposition by secreting degradative enzymes, further promoting the release and accumulation of DOC and EOC [48]. The synergistic effect of these three inputs not only increases the labile organic carbon fraction in the soil but also strengthens the stability of the soil carbon pool, making SOC more bioavailable and reducing carbon losses.

The application of bio-organic fertilizer also led to significant changes in soil enzyme activity. Compared to the treatment with chemical fertilizers alone, bio-organic fertilizer treatments resulted in a substantial increase in soil enzyme activity. This can be attributed to the fact that organic fertilizers provide a rich source of carbon and nutrients, which stimulate the growth and metabolism of soil microorganisms [49], consistent with results reported by [50]. Compared to the individual applications of cow manure and *Bacillus amyloliquefaciens* or mineral potassium fulvic acid, the combined application of cow manure, *Bacillus amyloliquefaciens,* and mineral potassium fulvic acid significantly increased soil urease activity. This suggests that the synergistic effects between cow manure, *Bacillus amyloliquefaciens,* and mineral potassium fulvic acid not only stimulated microbial populations but also enhanced the microbial processes essential for the efficient hydrolysis of urea, thereby improving nitrogen availability [51]. Applications of bio-organic fertilization can enhance microbial community stability, promote nutrient cycling, and contribute to soil organic matter accumulation, potentially improving soil fertility over multiple growing seasons. Additionally, bio-organic fertilization may reduce dependency on synthetic fertilizers by increasing nutrient-use efficiency, making it a more sustainable agricultural practice. However, long-term field studies are necessary to evaluate its cumulative effects on soil microbial diversity and nutrient availability. Future research should focus on assessing whether the observed benefits persist or diminish over time under different soil and climatic conditions.

### 4.2. Optimum Organic Fertilization Maintained Higher Soil Microbial Community Diversity and Ecosystem Multifunctionality

In intensively managed agricultural soils, the application of organic nutrients primarily accelerates nutrient cycling by increasing microbial biomass rather than significantly altering microbial community composition [52]. Our results partially support this finding. Specifically, compared to CK (NPK fertilization), various bio-organic fertilizer treatments enhanced soil bacterial richness (ACE and Chao indices) and diversity (Shannon index). However, fungal richness and diversity indices showed little change (Figure 4). This disparity is likely due to bacteria being more responsive to fertilization-induced changes than fungi [53]. These shifts can be attributed to the higher nutrient availability under bio-organic fertilizer treatments [54,55] and the formation of a more intricate bacterial network compared to inorganic fertilization [56]. The increase in soil bacterial diversity suggests that soil multifunctionality may be more resilient to environmental changes. Furthermore, PCA results indicated no significant differences in microbial community composition between CK and bio-organic fertilizer treatments (Figure 3). This may be because bio-organic fertilizers typically require long-term application to significantly alter microbial composition by enriching beneficial populations, increasing soil organic matter content, and modifying nutrient cycling processes [57]. Therefore, short-term studies may not fully capture the extent of microbial community changes. Additionally, microbial ecosystems often exhibit functional redundancy, where different microbial species perform similar ecological roles [58]. Even if bio-organic fertilizers alter microbial abundance at the species level, the overall community composition may remain unchanged due to functional substitutions among microbial taxa [59].

Microorganisms do not exist in isolation but instead form intricate networks of interactions [60]. The soil microbial co-occurrence network analysis indicates that greater microbial diversity contributes to higher network complexity within the community, leading to stronger connections between different microbial groups and enhancing the multifunctionality of soil ecosystems [61]. Compared to CK, bio-organic fertilization led to a more intricate bacterial network. This result is consistent with a recent study comparing organic and conventional intensive farming, which found that organic farming fosters greater microbial network complexity [62]. This increased complexity may make the ecosystem more resilient to environmental disturbances, as different microbial taxa can support and compensate for each other [59]. The reduction in bacterial network complexity under CK treatment may result from resource limitations that negatively affect microbial diversity and complexity. This finding aligns with observations linking conventional fertilization to relatively low soil organic carbon (SOC) levels, which contribute to reduced soil quality [63]. A highly complex microbial network enhances interactions among taxa, improving the multifunctionality of the soil ecosystem. This is particularly important for microbial communities that heavily depend on diverse metabolic pathways [64].

### 4.3. Effect of Soil Properties and Microbial Diversity on Ecosystem Multifunctionality

Soil microorganisms utilize a variety of substrates in different biological processes, thereby enhancing their functional diversity [65]. The ability of soil to sequester and release carbon is largely regulated by multiple soil biological processes [66]. Studies have shown that bacterial diversity plays a significant role in maintaining ecosystem multifunctionality, a conclusion further supported by random forest analysis, which aligns with the findings of [67]. However, compared to bacteria, fungal diversity contributes relatively less to ecosystem multifunctionality. This may be attributed to the following three factors: (1) fungi primarily decompose complex organic matter such as lignin and cellulose, and their growth is slower, contributing more significantly to the long-term stabilization of organic matter [68]. Consequently, fungal diversity may contribute less to ecosystem multifunctionality (EMF) in the short term. (2) Research has shown that bacterial communities are more sensitive to changes in soil soluble organic carbon, nitrogen, and other nutrients [69]. After organic fertilizer application, bacteria can rapidly decompose organic matter, promoting nitrogen mineralization and carbon cycling. Fungi, on the other hand, tend to dominate in nutrient-poor or complex organic matter environments, particularly in soils with long-term undisturbed conditions or high organic matter accumulation [70]. Therefore, the direct impact of organic fertilizer application on fungal communities is relatively small, and their contribution to ecosystem multifunctionality is more delayed. (3) Many fungi, such as mycorrhizal fungi, primarily influence ecosystem multifunctionality through interactions with plants (e.g., enhancing nutrient uptake) rather than directly regulating soil nutrient transformations [71]. Additionally, Tripathi et al. [72] suggested that soil pH is a primary driver of microbial diversity and ecosystem multifunctionality (EMF). However, our study did not confirm a significant effect of pH on microbial communities and EMF. This may be due to the relatively high soil pH in our study area and the minimal variation in pH across different fertilization regimes. Therefore, under stable pH conditions, changes in microbial diversity are more likely influenced by factors such as carbon source availability, nutrient levels, and microbial interactions, rather than being directly driven by pH.

## 5. Conclusions

Compared to CK, bio-organic fertilization in silage corn cultivation maintained a relatively higher yield by enhancing soil labile organic carbon and enzyme activity. Correspondingly, bio-organic fertilizers increased the richness and diversity of soil bacteria but had minimal effects on fungal diversity and abundance. Furthermore, random forest modeling and SEM analysis identified soil bacterial diversity as the primary driver and predictor of EMF. Overall, our findings emphasize the crucial role of soil bacterial diversity and its relationship with soil chemical properties in determining EMF. Bio-organic fertilizers exhibit significant potential in improving soil fertility, productivity, and ecosystem functionality, depending on their composition and soil nutrient status. In this study, the optimal fertilization strategy for silage corn was the combined application of cow manure, *Bacillus amyloliquefaciens*, and mineral-derived potassium fulvic acid, which fostered a healthier soil ecosystem characterized by higher corn yield and EMF values. Future research could expand the applicability of this study by evaluating the effects of fertilization strategies under different soil types, environmental conditions, and maize varieties. First, the focus should be on the effectiveness of fertilization strategies under various soil conditions, particularly differences in soil texture, organic matter content, pH, and salinity. Next, considering environmental factors such as temperature, precipitation patterns, and climate change, it is important to assess the long-term effects of bio-organic fertilizers under these variables. Finally, investigating the nutrient requirements and stress tolerance of different maize varieties would help develop more precise fertilization strategies, ultimately improving yields and soil health.

## Figures and Tables

**Figure 1 microorganisms-13-00959-f001:**
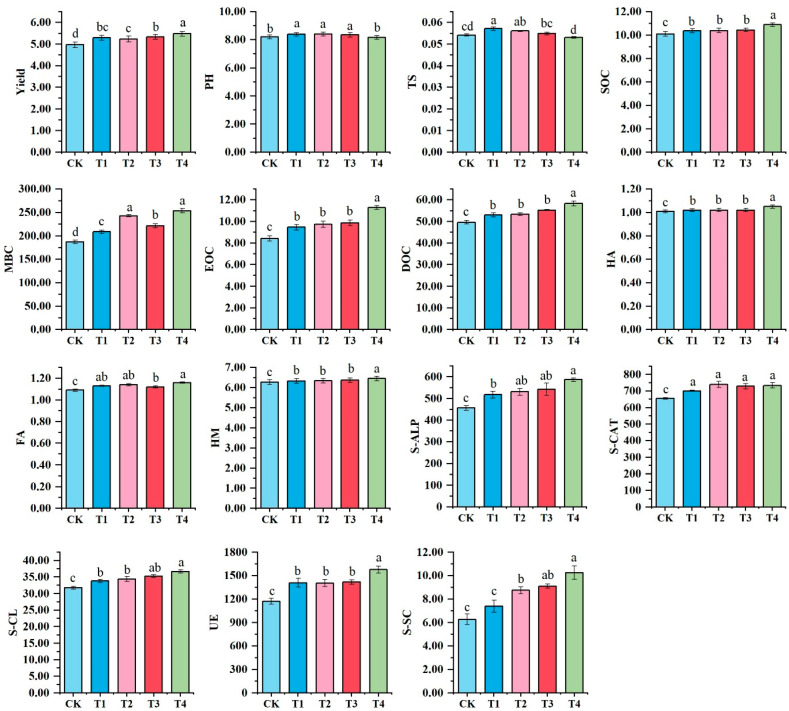
Soil physicochemical properties and microbiological properties in different organic fertilization treatments. Different letters indicated significant differences significantly among different fertilization treatments (Tukey’s multiple test, *p* < 0.05). TS: total salt; SOC: soil organic carbon; MBC: microbial biomass carbon; EOC: easily oxidized organic carbon; DOC: dissolved organic carbon; HA: humic acid; FA: fulvic acid; HM: humin; S-ALP: soil alkaline phosphatase; S-CAT: soil catalase; S-CL: soil cellulase; S-UE; soil urease; S-SC: soil sucrase. CK: chemical NPK fertilizer; T1: chemical NPK fertilizers (10% nitrogen reduction) plus well-decomposed cow manure; T2: chemical NPK fertilizers (10% nitrogen reduction) plus well-decomposed cow manure and *Bacillus amyloliquefaciens*; T3: chemical NPK fertilizers (10% nitrogen reduction) plus well-decomposed cow manure and mineral potassium fulvic acid; T4, chemical NPK fertilizers (10% nitrogen reduction) plus well-decomposed cow manure, *Bacillus amyloliquefaciens,* and mineral potassium fulvic acid.

**Figure 2 microorganisms-13-00959-f002:**
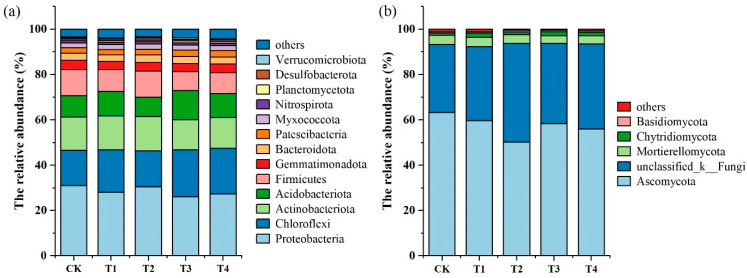
Relative abundance of bacterial and fungal phyla under different organic fertilization treatments. (**a**) Relative abundance of bacterial phyla and (**b**) relative abundance of fungal phyla.

**Figure 3 microorganisms-13-00959-f003:**
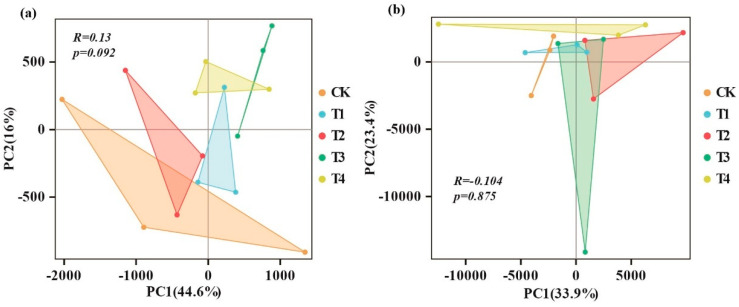
Principal component analysis (PCA) of bacterial and fungal communities under different organic fertilization treatments. (**a**) PCA of the bacterial community and (**b**) PCA of the fungal community.

**Figure 4 microorganisms-13-00959-f004:**
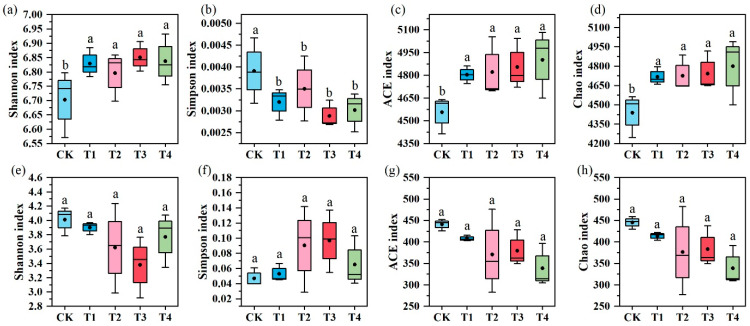
Soil microbial community diversity and richness. Panels (**a**,**b**) represent the bacterial community diversity indices (Shannon and Simpson), while panels (**c**,**d**) show the bacterial community richness indices (ACE and Chao). Panels (**e**,**f**) illustrate the fungal community diversity indices (Shannon and Simpson), and panels (**g**,**h**) are the fungal community richness indices (ACE and Chao). Different lowercase letters indicate significant differences with a *p* value < 0.05 based on the analysis of variance.

**Figure 5 microorganisms-13-00959-f005:**
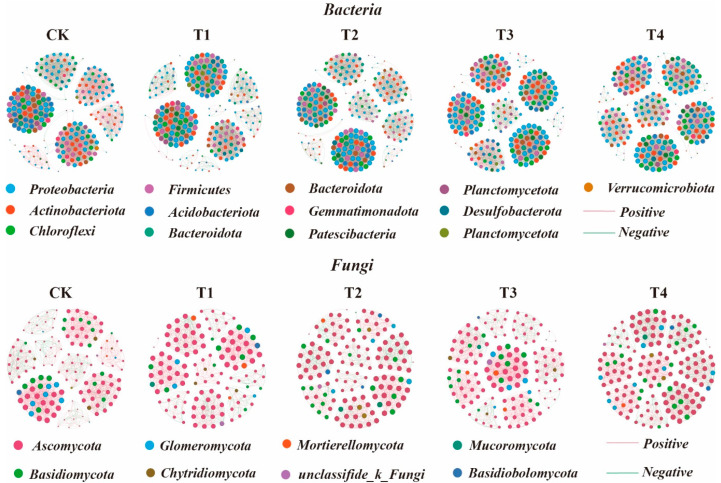
Soil microbial community co-occurrence networks under different organic fertilization treatments, based on Spearman correlations. Node size is proportional to the number of connections (degree), and edge thickness reflects the magnitude of Spearman’s correlation coefficients. Green edges represent negative interactions between bacterial nodes, while red edges represent positive interactions.

**Figure 6 microorganisms-13-00959-f006:**
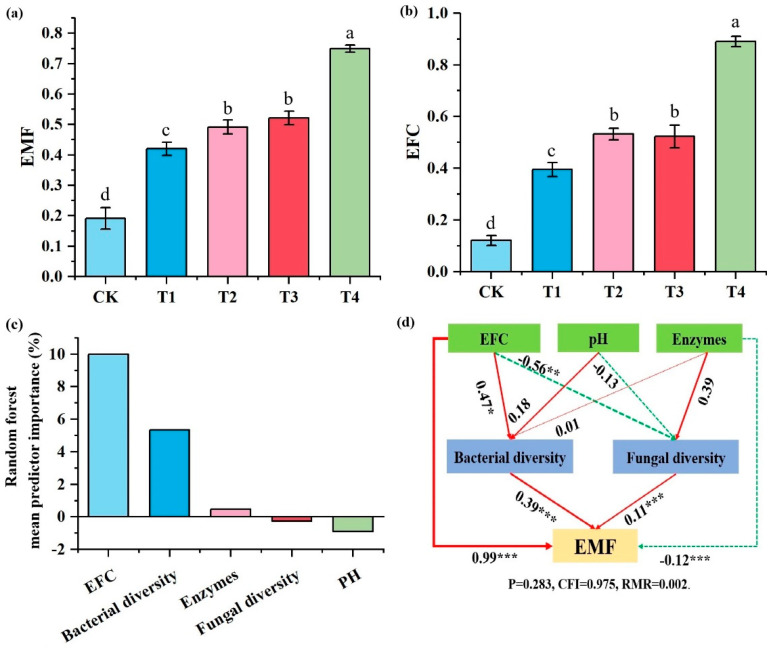
Effects of organic fertilization on ecosystem function and multifunctionality. (**a**) Soil carbon nutrients and storage function under different organic fertilization treatments. (**b**) Soil ecosystem multifunctionality under different organic fertilization treatments, where different lowercase letters indicate significant differences among treatments (*p* < 0.05). (**c**) Random forest analysis. (**d**) Structural equation models illustrating the effects of organic fertilization on ecosystem function and multifunctionality. Red solid arrows represent significant positive correlations, while green dashed arrows indicate negative correlations. The numbers on the arrows correspond to standardized path coefficients. Significance levels are denoted as * *p* < 0.05, ** *p* < 0.01, and *** *p* < 0.001.

**Figure 7 microorganisms-13-00959-f007:**
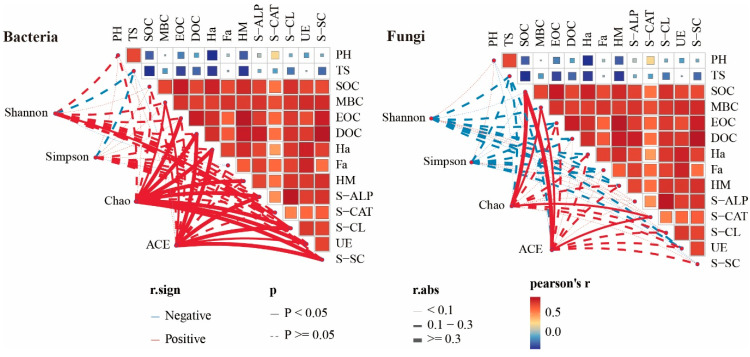
Mantel test for soil physicochemical properties and microbial diversity. Note: The width of the line indicates the magnitude of the correlation, the straight line represents significant, and the dotted line represents non-significant.

**Table 1 microorganisms-13-00959-t001:** Topological properties of soil microbial community co-occurrence networks.

Treatments	Bacteria					Fungi				
	CK	T1	T2	T3	T4	CK	T1	T2	T3	T4
Nodes	231	233	244	237	248	149	129	138	141	145
Links	3676	4056	4295	3956	4487	1341	925	965	1128	1236
Positive links	1922	2123	2204	2099	2584	946	729	708	945	1017
Negative links	1754	1933	2091	1857	1903	395	196	257	183	219
Positive links%	52.29	52.34	51.32	53.06	57.59	70.54	78.81	73.37	83.78	82.28
Average degree	31.827	36.053	35.205	33.384	39.17	16.868	14.341	13.986	16	17.048
Modularity	0.718	0.722	0.753	0.809	0.825	0.831	0.845	0.873	0.836	0.875
Grpah density	0.138	0.161	0.145	0.141	0.176	0.107	0.112	0.102	0.114	0.118

## Data Availability

The sequence data associated with this project have been deposited in the NCBI database under accession number PRJNA1234875.

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
