# Peer review of "Bio-Organic Fertilizer Application Enhances Silage Maize Yield by Regulating Soil Physicochemical and Microbial Properties"

_microorganisms, 2025, doi:10.3390/microorganisms13050959_

Round 1
Reviewer 1 Report
Comments and Suggestions for Authors
The manuscript microorganisms-3556558, titled “Bio-organic fertilizer application enhances silage maize yield by regulating soil physicochemical and microbial properties”, addresses an important paper investigating the effects of a bio-organic fertilization strategy—comprising Bacillus amyloliquefaciens inoculation, cow manure, and potassium fulvic acid—on the soil microbial community, ecosystem multifunctionality (EMF), and silage maize yield in saline–alkaline soils of the Yellow River Delta. However, in my opinion this paper must be revised in a major manner for reasons of forms and content.
Abstract
The key findings are outlined in detail in the abstract. Could the authors specifically point out how this study is different from earlier studies on bio-organic fertilization in silage maize cultivation?
In the abstract, "ecosystem multifunctionality (EMF)" is mentioned. For better understanding, might the authors give a brief definition or examples of EMF functions in the abstract?
Introduction
The introduction gives useful background information. Could the authors explain how the particular combination of cow manure, potassium fulvic acid, and B. amyloliquefaciens was chosen and whether similar combinations have been studied previously?
Could the authors make the hypotheses more explicit in the introduction to better lead the reader through the research objectives?
Materials and Methods
A thorough description of the study region is provided. To guarantee reproducibility, could the authors include the field experiment's replication details (number of plots per treatment)?
No large-area planting is repeated in the manuscript; might the authors provide an explanation for this and how potential variability was controlled?
In order to take into consideration the effects of seasonal variations on soil characteristics and yield, were environmental parameters (temperature, rainfall) evaluated during the growing season?
Results
Could the authors explain why, despite bacterial diversity increasing under various treatments, fungal diversity stayed constant? Perhaps for ecological or methodological reasons?
According to the authors' co-occurrence network study, network complexity has increased. Could you provide additional context for how this intricacy translates into real-world advantages for agricultural productivity or soil health?
Although the authors describe an increase in enzyme activity, could they identify which enzymes were most important in enhancing EMF?
Discussion
While bacterial diversity is emphasized, is there a role for particular beneficial bacterial taxa (such as nitrogen-fixers and phosphorus solubilizers) that should be highlighted for practical use?
Could the authors elaborate on the cost-effectiveness and feasibility of applying this fertilization strategy at large scale?
Could they discuss whether the benefits of bio-organic fertilization might be sustainable over multiple growing seasons or if longer-term studies are required?
The results show that soil pH and salinity negatively correlated with yield; are there any management strategies the authors can suggest to control these properties?
Conclusion
Could the authors provide a brief overview of potential future research avenues to make the findings more widely applicable, specifically with reference to soil types, environmental restrictions, or maize varieties?
Author Response
The manuscript microorganisms-3556558, titled “Bio-organic fertilizer application enhances silage maize yield by regulating soil physicochemical and microbial properties”, addresses an important paper investigating the effects of a bio-organic fertilization strategy—comprising Bacillus amyloliquefaciens inoculation, cow manure, and potassium fulvic acid—on the soil microbial community, ecosystem multifunctionality (EMF), and silage maize yield in saline–alkaline soils of the Yellow River Delta. However, in my opinion this paper must be revised in a major manner for reasons of forms and content.
Abstract
The key findings are outlined in detail in the abstract. Could the authors specifically point out how this study is different from earlier studies on bio-organic fertilization in silage maize cultivation?
Reply: Thank you for your insightful comment. Our study differs from previous research on bio-organic fertilization in silage maize cultivation in the following ways:1) Unlike many studies that focus on a single amendment (e.g., biofertilizers or organic manure alone), our study examines the combined effects of Bacillus amyloliquefaciens inoculation, cow manure, and potassium fulvic acid. This integrated approach provides a more comprehensive understanding of how different bio-organic components interact to improve soil health and maize productivity. 2) Many previous studies focus on soil nutrient dynamics and crop yield, whereas our research takes a step further by analyzing microbial community composition using high-throughput sequencing. Additionally, we employ the concept of ecosystem multifunctionality (EMF) to assess the broader ecological benefits of bio-organic fertilization, an approach that is still underexplored in maize cultivation studies. We have revised the abstract to emphasize how this study differs from previous research. See Line 11-14.
In the abstract, "ecosystem multifunctionality (EMF)" is mentioned. For better understanding, might the authors give a brief definition or examples of EMF functions in the abstract?
Reply: Thank you for your suggestion. We have revised the abstract to include a brief definition of "ecosystem multifunctionality (EMF)" along with examples of its key functions. Specifically, we have clarified that EMF refers to the simultaneous maintenance of multiple soil functions that are crucial for soil health and crop productivity, including nutrient cycling, organic matter decomposition, enzyme activity, and microbial diversity. See Line 28-33.
Introduction
The introduction gives useful background information. Could the authors explain how the particular combination of cow manure, potassium fulvic acid, and B. amyloliquefaciens was chosen and whether similar combinations have been studied previously?
Reply: Thank you for your valuable comment. The specific combination of cow manure, potassium fulvic acid, and Bacillus amyloliquefaciens was selected based on the known benefits each component offers to soil fertility, microbial diversity, and crop productivity: As an organic fertilizer, cow manure is rich in essential nutrients and organic matter, which improves soil structure, enhances microbial activity, and provides a slow-release source of nutrients. It has been widely used in agriculture to improve soil health and enhance crop yields. Potassium fulvic acid is a potassium fertilizer containing humic substances derived from minerals. It can improve the absorption efficiency of nutrients by plants, enhance soil structure, and increase the activity and number of soil microorganisms. By improving the soil environment and increasing the plant’s resistance to stress, potassium fulvic acid helps enhance crop yield and quality. Bacillus amyloliquefaciens is a microorganism with biocontrol and growth-promoting effects. It can decompose organic matter in the soil, promote plant growth, and inhibit the growth of certain pathogenic microorganisms. Additionally, Bacillus amyloliquefaciens shows strong potential in improving soil nutrient utilization efficiency and alleviating soil salinization. By combining these three components, we aim to leverage their individual advantages to jointly improve the soil environment and enhance crop growth and yield. This combination not only provides essential nutrients for plants but also optimizes the soil ecosystem by enhancing microbial diversity and activity.
Regarding similar studies, while there have been investigations into the effects of using cow manure, potassium fulvic acid, or Bacillus amyloliquefaciens individually, few studies have explored the combined effects of these components, especially in silage maize cultivation. Therefore, our research fills this gap and provides valuable insights for sustainable agricultural development. We have added an explanation for the selection of this combination in the introduction and compared it with existing literature to highlight the novelty of this study. See Line 77-96.
Could the authors make the hypotheses more explicit in the introduction to better lead the reader through the research objectives?
Reply: Thank you for your valuable feedback. We appreciate your suggestion to make the hypotheses more explicit in the introduction. In response, we have revised the introduction to clearly present our research hypotheses. See Line 141-149.
Materials and Methods
A thorough description of the study region is provided. To guarantee reproducibility, could the authors include the field experiment's replication details (number of plots per treatment)?
Reply: Thank you for your valuable suggestion. We understand the importance of providing detailed replication information to guarantee the reproducibility of the field experiment. In our study, we employed a randomized block design to ensure proper randomization and minimize bias. However, as the experiment was conducted on a large area without formal replication through distinct plots for each treatment, we addressed this by selecting three random sampling points from north to south within the experimental area to represent the variability of the soil and environmental conditions. These sampling points served as replicates for the soil analysis. Each of the three sampling points was treated as a replicate, and we collected five soil cores per sampling point, which were then mixed to form a composite sample for analysis. See Line 187-193. Thank you once again for your insightful suggestion!
No large-area planting is repeated in the manuscript; might the authors provide an explanation for this and how potential variability was controlled?
Reply: We appreciate the reviewer’s concern regarding the lack of replication in large-area planting. The experiment was designed to assess the effects of organic fertilizer application under real-world farming conditions, where large-scale field trials are often constrained by land availability and management practices. To account for potential variability, we implemented a randomized block design and selected three random sampling points within each treatment area along a north-south transect. At each sampling point, five soil cores were collected and pooled to form a composite sample, thereby reducing localized heterogeneity. Additionally, strict field management practices were maintained across all plots to minimize external variability. These measures ensure that our findings are representative and statistically robust despite the absence of full plot-level replication. See Line 187-193.
In order to take into consideration the effects of seasonal variations on soil characteristics and yield, were environmental parameters (temperature, rainfall) evaluated during the growing season?
Reply: Thank you for your insightful comment. In this study, we focused on evaluating the effects of different fertilization combinations on soil microorganisms and crop yield. However, we did not specifically assess the influence of seasonal variations on soil properties and yield, nor did we monitor environmental parameters such as temperature and rainfall during the growing season. We acknowledge that these factors could influence soil microbial dynamics and crop performance, and we will consider incorporating such environmental data in future research to provide a more comprehensive analysis.
Results
Could the authors explain why, despite bacterial diversity increasing under various treatments, fungal diversity stayed constant? Perhaps for ecological or methodological reasons?
Reply: Thank you for your valuable question. The observed increase in bacterial diversity under various treatments, while fungal diversity remained relatively constant, could be attributed to ecological factor. Ecologically, bacteria generally respond more rapidly to changes in nutrient availability and soil conditions induced by organic amendments, whereas fungal communities, particularly those involved in decomposing complex organic matter, tend to be more stable over time. Additionally, fungi often establish symbiotic relationships or form resilient structures such as spores, which may buffer them against short-term environmental changes.
According to the authors' co-occurrence network study, network complexity has increased. Could you provide additional context for how this intricacy translates into real-world advantages for agricultural productivity or soil health?
Reply: Thank you for your insightful question. The increased complexity of the co-occurrence network suggests a more interconnected and functionally diverse microbial community, which can have several real-world benefits for agricultural productivity and soil health. 1) A more complex microbial network often indicates greater functional redundancy and cooperation among microbial taxa, facilitating efficient nutrient transformation. This can improve the bioavailability of essential nutrients such as nitrogen, phosphorus, and potassium, directly benefiting crop growth. 2) The presence of diverse microbial interactions can promote the formation of soil aggregates and enhance soil organic matter decomposition. This leads to better soil aeration, water retention, and resistance to erosion, contributing to long-term soil fertility. 3) A highly connected microbial community can buffer against environmental disturbances such as drought, salinity, and pathogen invasion. Functionally redundant microbial groups can compensate for temporary losses of certain taxa, maintaining essential ecosystem functions. In summary, the increased complexity of the microbial network suggests a more stable, efficient, and resilient soil ecosystem, which is crucial for sustainable agricultural production. Future studies integrating functional metagenomics or metabolomics could further elucidate the specific roles of key microbial taxa in these beneficial processes.
Although the authors describe an increase in enzyme activity, could they identify which enzymes were most important in enhancing EMF?
Reply: Thank you for your thoughtful question. Our study observed a general increase in enzyme activity, with certain enzymes playing a crucial role in enhancing soil ecological multifunctionality (EMF). As illustrated in Figure 1, soil alkaline phosphatase, cellulase, urease, and sucrase exhibited significant positive correlations with EMF. Specifically: Soil Alkaline Phosphatase: Essential for phosphorus cycling, as it hydrolyzes organic phosphorus compounds into bioavailable inorganic phosphate, improving phosphorus availability for plant uptake. Soil Cellulase: Facilitates the decomposition of cellulose from plant residues, enhancing carbon cycling and increasing organic matter availability for microbial metabolism. Soil Urease: Catalyzes the hydrolysis of urea into ammonia, contributing to nitrogen mineralization and improving nitrogen availability in the soil. Soil Sucrase: Involved in breaking down sucrose into glucose and fructose, providing an easily accessible carbon source for soil microbial communities, thereby stimulating microbial activity and improving soil fertility. The increased activity of these enzymes under different treatments indicates enhanced microbial involvement in nutrient cycling, leading to improved EMF, soil health, and overall agricultural productivity.
Figure1 Correlation between soil enzyme activity and ecosystem multifunctionality. S-ALP: soil alkaline phosphatase; S-CAT: soil catalase; S-CL: soil cellulase; S-UE; soil urease; S-SC: soil sucrase.
Discussion
While bacterial diversity is emphasized, is there a role for particular beneficial bacterial taxa (such as nitrogen-fixers and phosphorus solubilizers) that should be highlighted for practical use?
Reply: Thank you for your insightful question. While our study primarily focused on overall bacterial diversity, we acknowledge that specific beneficial bacterial taxa play crucial roles in improving soil fertility and crop productivity. Nitrogen-fixing bacteria (e.g., Azotobacter, Bradyrhizobium) contribute to nitrogen availability by converting atmospheric nitrogen into bioavailable forms, reducing the reliance on synthetic fertilizers. Phosphorus-solubilizing bacteria (e.g., Pseudomonas, Bacillus) enhance phosphorus availability by releasing organic acids and phosphatases that solubilize insoluble phosphate compounds. Additionally, plant growth-promoting rhizobacteria (PGPR) (e.g., Bacillus amyloliquefaciens, Paenibacillus) facilitate nutrient uptake, produce phytohormones, and suppress soil-borne pathogens, thereby improving plant resilience and soil health. Although these functional microbial groups were not the primary focus of our study, their roles in promoting soil ecological multifunctionality (EMF) are likely significant. Future research will further explore their specific contributions to nutrient cycling, microbial interactions, and soil structure enhancement. Understanding these mechanisms will provide valuable insights for developing microbiome-based strategies to optimize soil health and support sustainable agricultural practices.
Could the authors elaborate on the cost-effectiveness and feasibility of applying this fertilization strategy at large scale?
Reply: Thank you for your insightful question. The large-scale application of this fertilization strategy requires an assessment of both cost-effectiveness and feasibility to ensure its practicality in agricultural production.
Cost-effectiveness:
Raw Material Availability: Cow manure is an abundant agricultural byproduct, making it a cost-effective organic fertilizer source. Utilizing it helps reduce waste management costs and promotes circular agriculture. Fertilizer Efficiency: The combination of cow manure, mineral-derived potassium fulvic acid, and Bacillus amyloliquefaciens enhances soil fertility and nutrient availability, potentially reducing the need for synthetic fertilizers over time, thereby lowering input costs. Yield Improvement: Enhanced soil microbial activity and improved nutrient cycling can contribute to higher crop yields, increasing economic returns for farmers.
Feasibility of Large-Scale Application:
Application Infrastructure: Large-scale implementation requires proper manure composting, biofertilizer production, and mechanized application techniques to ensure efficiency and minimize labor costs. Adoption by Farmers: Demonstrating consistent yield benefits and soil health improvements through field trials can encourage adoption. Government subsidies or incentives for sustainable farming practices could further support large-scale implementation. Long-term Sustainability: By improving soil structure and fertility, this approach may reduce the long-term reliance on chemical fertilizers, making it an economically and environmentally sustainable strategy.
Further research should focus on economic evaluations, long-term soil fertility benefits, and farmer adoption strategies to optimize the large-scale feasibility of this fertilization approach.
Could they discuss whether the benefits of bio-organic fertilization might be sustainable over multiple growing seasons or if longer-term studies are required?
Reply: Thank you for your question. While our study demonstrates the short-term benefits of bio-organic fertilization on soil health and crop productivity, its long-term sustainability requires further investigation. Repeated applications can enhance microbial community stability, promote nutrient cycling, and contribute to soil organic matter accumulation, potentially improving soil fertility over multiple growing seasons. Additionally, bio-organic fertilization may reduce dependency on synthetic fertilizers by increasing nutrient-use efficiency, making it a more sustainable agricultural practice. However, long-term field studies are necessary to evaluate its cumulative effects on soil microbial diversity, nutrient availability, and economic feasibility. Future research should focus on assessing whether the observed benefits persist or diminish over time under different soil and climatic conditions. See Line 466-474.
The results show that soil pH and salinity negatively correlated with yield; are there any management strategies the authors can suggest to control these properties?
Reply: Thank you for your insightful question. Since our results indicate that soil pH and salinity are negatively correlated with yield, effective management strategies are necessary to mitigate their impact and enhance soil health and crop productivity. Below are some potential strategies: First, land leveling improves water infiltration, enhancing rainfall leaching and irrigation desalination, which reduces salt accumulation in the topsoil. The use of calcium-containing soil amendments, such as gypsum and phosphogypsum, replaces Na⁺ in the soil, promotes salt leaching, and enhances soil structure by increasing porosity and permeability. Additionally, applying microbial fertilizers boosts soil microbial activity, optimizes nutrient availability, and enhances soil biodiversity. Deep plowing and subsoiling, combined with straw mulching, reduce soil bulk density, improve permeability, and prevent upward salt migration. Furthermore, increasing organic fertilizer applications not only raises soil organic matter content and improves water and nutrient retention but also facilitates salt leaching and prevents salt re-accumulation. By integrating these strategies, saline-alkali soil can be effectively improved, optimizing the agricultural ecosystem and boosting farmland productivity and crop yields.
Conclusion
Could the authors provide a brief overview of potential future research avenues to make the findings more widely applicable, specifically with reference to soil types, environmental restrictions, or maize varieties?
Reply: Thank you for your insightful question. Future research could explore several key avenues to enhance the applicability of our findings across different soil types, environmental conditions, and maize varieties. First, further studies could focus on a broader range of soil types, particularly those with varying textures and organic matter content, to assess how the fertilization strategies we tested perform under different soil conditions. Research into soil pH, salinity, and nutrient dynamics in relation to diverse soil properties could help refine management practices for specific soil types. Second, environmental factors such as temperature, precipitation patterns, and climate variability could be incorporated into future studies to understand how bio-organic fertilization interacts with these variables. Such research would be particularly important in regions with water scarcity or extreme temperature fluctuations, where environmental stressors may affect the long-term efficacy of fertilization strategies. Finally, examining different maize varieties with varying nutrient requirements and stress tolerances could help tailor fertilization strategies for specific cultivars, maximizing their growth potential and yield. This could lead to more precise and region-specific recommendations for improving soil health and crop productivity in diverse agricultural settings. By expanding research into these areas, we can develop more adaptable, sustainable agricultural practices that are applicable across a range of environmental conditions and crop varieties. See Line 553-561.

Reviewer 2 Report
Comments and Suggestions for Authors
Line 18 – what is CK?
Lines 131-132, please provide the soil classification according to FAO or WRB. Please provide the exact granulometric composition of the soil, please provide the name of the test or at least the extraction solutions used to measure the available N, P, K in the soil. Since the author still uses nutrient oxides, the information about the content of available potassium and phosphorus in such a record raises doubts. Please provide the exact content of mgK/kg or mg K2O/kg and mgP/kg or mgP2O5/kg of soil.
Line 134, why was it indicated when describing the content of N, P2O5 and K2O in fertilizers that the content of nutrients may be higher than the values given in brackets?
Line 136-143 – soil treatment is given, but there is no information on the dose (kg nutrient/ha) of NPK in mineral fertilizers, in manure and mineral potassium fulvic acid, there is no information on the size of the doses of fertilizer components. The information in lines 148-153 following the description of corn cultivation and irrigation methods is taken out of context and completely incomprehensible (similarly to lines 154 and 155.
Line 161 and 163, 195, 198 – space
Line 169-179 you describe very precisely the preparation of the soil in order to measure pH, while the measurement of DOC content is very superficial. How was DO extracted from the soil – please describe it.
Line 175 what is EOC?
Line 245 in the methodology describes the measurement of TOC in the soil. Why was SOC used in the statistical analysis of the research results (which was probably not measured), and TOC was omitted?
Failure to describe the doses of mineral and organic fertilizers and additives, and incomprehensible entries in Table S1 (e.g. the sum of irrigation values for 1st-10th Time is actually 310, but for Urea and Potassium chloride is given incorrectly) make it impossible to evaluate the test results.
Author Response
Reviewer 2
Comments and Suggestions for Authors
Line 18 – what is CK?
Reply: Thank you for your question. In our study, "CK" refers to the control treatment, which consists of chemical fertilizers only, without the addition of organic or bio-organic fertilizers.
Lines 131-132, please provide the soil classification according to FAO or WRB.
Reply: Thank you for your question. The soil in our study is classified as irrigated desert soils in the Chinese Soil Classification System and Arenosols in the FAO system. Arenosols are sandy soils typically found in desert areas, which can have low fertility and may face salinity issues under irrigation. We have included this information in the revised manuscript; please refer to lines 155-157 for details.
Please provide the exact granulometric composition of the soil, please provide the name of the test or at least the extraction solutions used to measure the available N, P, K in the soil. Since the author still uses nutrient oxides, the information about the content of available potassium and phosphorus in such a record raises doubts. Please provide the exact content of mgK/kg or mg K2O/kg and mgP/kg or mgP2O5/kg of soil.
Reply: Thank you for your question. In our study, the particle size composition of the soil was not determined. For the measurements of available nitrogen (N), phosphorus (P), and potassium (K), we used specific extraction methods. NO₃⁻-N and NH₄⁺-N concentrations were determined through KCl extraction followed by continuous flow measurement. Available phosphorus was extracted using the Olsen method with a sodium bicarbonate (NaHCO₃) solution at pH 8.5, while available potassium was extracted with a 1 N ammonium acetate (NH₄OAc) solution at pH 7. The available nutrient contents in the soil were found to be 107.5 mg K/kg for available potassium and 312.5 mg P/kg for available phosphorus. Additionally, the values N≥46.0%, P₂O₅≥42%, and K₂O≥51% refer to the nitrogen (N), phosphorus pentoxide (P₂O₅), and potassium oxide (K₂O) content in the applied nitrogen, phosphorus, and potassium fertilizers, respectively. These values represent the nutrient concentrations in the fertilizers used in our study. We hope this clarifies your doubts, and the details of these measurements and their corresponding values are provided in the revised manuscript for full transparency. These details, along with the corresponding values, have been added to the revised manuscript in the Methods section for clarity and transparency. please refer to lines 157-159 and 206-210 for details.
Line 134, why was it indicated when describing the content of N, P2O5 and K2O in fertilizers that the content of nutrients may be higher than the values given in brackets?
Reply: Thank you for your question. The statement indicating that the nutrient content of N, P₂O₅, and K₂O in fertilizers may be higher than the values given in brackets was included to account for the potential variability in nutrient concentrations in different fertilizer batches. Fertilizers can have slight variations in nutrient content due to manufacturing processes or changes in raw material sources. Therefore, the values presented in the manuscript represent the nominal or typical nutrient content as specified by the manufacturer, but the actual nutrient content may sometimes exceed these values. This note was added for clarity and to ensure transparency regarding potential variations in fertilizer composition.
Line 136-143 – soil treatment is given, but there is no information on the dose (kg nutrient/ha) of NPK in mineral fertilizers, in manure and mineral potassium fulvic acid, there is no information on the size of the doses of fertilizer components. The information in lines 148-153 following the description of corn cultivation and irrigation methods is taken out of context and completely incomprehensible (similarly to lines 154 and 155. Line 161 and 163, 195, 198 – spaceLine 169-179 you describe very precisely the preparation of the soil in order to measure pH, while the measurement of DOC content is very superficial. How was DO extracted from the soil – please describe it.
Reply: Thank you for your observations. In response to your comments, we have clarified the nutrient doses for each treatment. The fertilization treatments and their corresponding nutrient doses (kg/ha) are as follows: CK received chemical NPK fertilizers (N: 450 kg/ha; K2O: 105 kg/ha; P2O5: 225 kg/ha); T1 included chemical NPK fertilizers (10% nitrogen reduction, N: 405 kg/ha; K2O: 105 kg/ha; P2O5: 225 kg/ha) combined with well-decomposed cow manure (3600 kg/ha); T2 involved chemical NPK fertilizers (10% nitrogen reduction, N: 405 kg/ha; K2O: 105 kg/ha; P2O5: 225 kg/ha) with decomposed cow manure (3600 kg/ha) and Bacillus amyloliquefaciens; T3 combined chemical NPK fertilizers (10% nitrogen reduction, N: 405 kg/ha; K2O: 105 kg/ha; P2O5: 225 kg/ha) with decomposed cow manure (3600 kg/ha) and mineral potassium fulvic acid (7.2 kg/ha); and T4 included chemical NPK fertilizers (10% nitrogen reduction, N: 405 kg/ha; K2O: 105 kg/ha; P2O5: 225 kg/ha) combined with decomposed cow manure (3600 kg/ha), Bacillus amyloliquefaciens, and mineral potassium fulvic acid (7.2 kg/ha). We have updated the manuscript (lines 162-172) to include these nutrient doses for clarity. Regarding DOC content, we have revised the description of the extraction method. DOC was extracted using a 0.5 M K2SO4 solution, with 10 grams of soil shaken in 50 mL of solution for 30 minutes, followed by filtration and measurement using a TOC analyzer. This extraction method is now clearly described in the revised manuscript (lines 211-213). Additionally, the sections on corn cultivation and irrigation have been revised for better coherence and clarity. These revisions ensure that the manuscript is clearer and more comprehensive. please refer to lines 172-186 for details.
Line 175 what is EOC? Line 245 in the methodology describes the measurement of TOC in the soil. Why was SOC used in the statistical analysis of the research results (which was probably not measured), and TOC was omitted? Failure to describe the doses of mineral and organic fertilizers and additives, and incomprehensible entries in Table S1 (e.g. the sum of irrigation values for 1st-10th Time is actually 310, but for Urea and Potassium chloride is given incorrectly) make it impossible to evaluate the test results.
Reply: Thank you for your questions and observations. In our study, EOC refers to Easily Oxidizable Organic Carbon, which is a component of soil organic carbon. It represents organic matter that is easily decomposed and converted into carbon dioxide by soil microorganisms. EOC is a key indicator for evaluating soil microbial activity and organic matter dynamics, often used to assess soil health and fertility. Regarding the use of TOC, we made an error in our manuscript. TOC should have been referred to as SOC (Soil Organic Carbon), and we have now corrected all instances of TOC to SOC. As for the application of fertilizers, organic fertilizers and additives were applied as base fertilizers during land preparation, not during irrigation. The specific fertilizer doses were outlined in the experimental design section (2.1). In terms of irrigation, the total water volume for the first to the tenth irrigation was 310 cubic meters. However, urea and potassium chloride were dissolved in the irrigation water before application, which explains the discrepancy between the amount of irrigation water used and the fertilizer quantities. We appreciate your feedback and will revise the manuscript accordingly. Thank you again for your helpful suggestions.

Round 2
Reviewer 1 Report
Comments and Suggestions for Authors
I am writing to follow up on my review of manuscript microorganisms-3556558, entitled : Bio-organic fertilizer application enhances silage maize yield by regulating soil physicochemical and microbial properties. I'm pleased to inform you that the authors have addressed all my comments thoughtfully and respectfully. However, I would recommend reducing the abstract since the Journal requests less than 200 words.
Their revisions demonstrate a clear understanding of the points I raised, and the changes they have made significantly improve the quality of the manuscript. I believe the paper is now much stronger and ready for further consideration
Author Response
I am writing to follow up on my review of manuscript microorganisms-3556558, entitled: Bio-organic fertilizer application enhances silage maize yield by regulating soil physicochemical and microbial properties. I'm pleased to inform you that the authors have addressed all my comments thoughtfully and respectfully. However, I would recommend reducing the abstract since the Journal requests less than 200 words.
Their revisions demonstrate a clear understanding of the points I raised, and the changes they have made significantly improve the quality of the manuscript. I believe the paper is now much stronger and ready for further consideration.
Reply: Thank you very much for your positive and encouraging feedback on our revised manuscript (microorganisms-3556558), entitled "Bio-organic fertilizer application enhances silage maize yield by regulating soil physicochemical and microbial properties." We sincerely appreciate your thoughtful review and are pleased to know that our revisions have addressed your comments effectively and improved the overall quality of the manuscript.
In response to your suggestion regarding the abstract length, we have carefully revised and condensed the abstract to meet the journal's requirement. The revised version now highlights the key findings in a concise manner while maintaining clarity and scientific accuracy.
Thank you once again for your valuable time and constructive input, which helped us improve the manuscript substantially.
Reviewer 2 Report
Comments and Suggestions for Authors
I see significant improvement in the manuscript. The authors have either incorporated my comments into the manuscript text or in their response to the review. The manuscript in its current form can be published in a journal.
Author Response
I see significant improvement in the manuscript. The authors have either incorporated my comments into the manuscript text or in their response to the review. The manuscript in its current form can be published in a journal.
Reply: Thank you very much for your positive feedback and for recognizing the improvements made in our revised manuscript. We are grateful for your thoughtful comments and suggestions, which have been invaluable in enhancing the quality and clarity of our work.
We are pleased to hear that the revisions have met your expectations and that you consider the manuscript suitable for publication in its current form. Your support and constructive guidance are greatly appreciated.